# Impact of correcting misinformation on social disruption

**Ryusuke Iizuka**[1]*, **Fujio Toriumi**[1], **Mao Nishiguchi**[1], **Masanori Takano**[2],
**Mitsuo Yoshida**[3]

1 The University of Tokyo, Tokyo, Japan, 2 CyberAgent, Inc., Tokyo, Japan, 3 Toyohashi University of
Technology, Toyohashi, Aichi, Japan

* iizuka@torilab.net

pone.0265734

**Data Availability Statement:** Regarding the data
about the tweets, we attached the TweetID. Data
about sales index cannot be shared publicly
because we purchased the data from NOWCAST,
INC. Therefore, we have special access privileges

## Abstract

People are obtaining more and more information from social media and other online
sources, but the spread of misinformation can lead to social disruption. In particular, social
networking services (SNSs) can easily spread information of uncertain authenticity and fac-
tuality. Although many studies have proposed methods that addressed how to suppress the
spread of misinformation on SNSs, few works have examined the impact on society of dif-
fusing both misinformation and its corrective information. This study models the effects of
effort to reduce misinformation and the diffusion of corrective information on social disrup-
tion, and it clarifies these effects. With the aim of reducing the impact on social disruption,
we show that not only misinformation but also corrective information can cause social dis-
ruption, and we clarify how to control the spread of the latter to limit its impact. We analyzed
the misinformation about a toilet-paper shortage and its correction as well as the social dis-
ruption this event caused in Japan during the COVID-19 pandemic in 2020. First, (1) we
analyzed the extent to which misinformation and its corrections spread on SNS, and then (2)
we created a model to estimate the impact of misinformation and its corrections on the
world. Finally, (3) We used our model to analyze the change in this impact when the diffusion
of the misinformation and its corrections changed. Based on our analysis results in (1), the
corrective information spread much more widely than the misinformation. From the model
developed in (2), the corrective information caused excessive purchasing behavior. The
analysis results in (3) show that the amount of corrective information required to minimize
the societal impact depends on the amount of misinformation diffusion. Most previous stud-
ies concentrated on the impact of corrective information on attitudes toward misinformation.
On the other hand, the most significant contribution of this study is that it focuses on the
impact of corrective information on society and clarifies the appropriate amount of it.

## Introduction

Social network services (SNSs) such as Twitter and Facebook are used by hundreds of millions
of people worldwide for information gathering and communication [1]. Thus, the diffusion of
information by SNSs significantly impacts society [2–4]. One example is the Arab Spring [5],

to the data. For more information, please contact us or NOWCAST, INC(info@nowcast.co.jp).

**Funding:** Masanori Takano received funding in the form of salary from CyberAgent, Inc during the course of the study. JSPS Grants-in-Aid for Scientific Research Grant Numbers 19H02376, and JST RISTEX Grant Number JPMJRX20J3, Japan. The specific roles of these authors are articulated in the 'author contributions' section. The funders had no role in study design, data collection and analysis, decision to publish, or preparation of the manuscript.

**Competing interests:** Masanori Takano is an employee of CyberAgent, Inc. There are no patents, products in development or marketed products associated with this research to declare. This does not alter our adherence to PLOS ONE policies on sharing data and materials.

which was a movement demanding democracy and freedom that emerged in the Arab world between 2010 and 2012. Conversations on social media deepened political discussions, and social media hugely influenced this movement [5]. "The Ice Bucket Challenge" of 2014 is another type of mass event created by social media [6]. This campaign, which supported amyotrophic lateral sclerosis (ALS) research, offered celebrities the choice of getting soaked by a bucket of ice water or donating to the ALS Association of America. Ice Bucket Challenge videos were viewed by 440 million people, and the campaign raised $220 million from 28 million people [6].

Unfortunately, the diffusion of information on SNSs can also create negative outcomes, such as misinformation [7]. The spread of unreliable information can lead to social disruption and even cause serious problems [8, 9]. In this study, we define misinformation as information that contains errors, regardless of whether those spreading it actually intend to misinform others. According to Ten Brinke et al. [10], the definition of social disruption is a situation in which the continuity is disturbed of processes that are vital to the functioning of society. These processes refer to both physical and socio-psychological aspects. Socio-psychological aspects are disruption of daily life, loss of trust in authorities, not knowing what to do, fear, etc. In this paper, we focus on socio-psychological aspects and define social disruption as a situation that people do not know what to do due to fake news.

In the 2016 U.S. presidential election, a great deal of fake news was spread, and many people believed it [8]. The most widely spread fake news favored Donald Trump, not Hillary Clinton, and the most common type of fake news favored Trump, not Clinton. These facts suggest that the election's outcome might have been different without such fake news. The so-called "Pizzagate" conspiracy theory is another example of the impact of misinformation on SNS. It refers to a shooting incident at a pizza restaurant in Washington, D.C., on December 4, 2016. The shooter believed this conspiracy theory that was spread on SNS, and he admitted that his violent reactions were influenced by it [9].

One way to prevent the spread of misinformation is to reduce the number of people who believe it by providing corrective information. According to M.S. Chan et al. [11], when there is a strong reason to believe misinformation, the effect of corrective information on people's misperceptions is weakened, and thus more detailed corrective information is needed to more effectively counter people's misperceptions. Corrective information can in fact be counterproductive and fail to dissuade those who do believe the falsehoods. This is called the backfire effect [12]. In this phenomenon, when people encounter information they do not want to believe, they do not change their original beliefs but instead even they strengthen them. Another method detects accounts that spread misinformation and then prevents those account's owners from engaging in such dissemination. In particular, since bots account for a large percentage of the accounts that spread fake news [13], many studies have concentrated on methods that detect malicious bots [14, 15].

Therefore, many studies have addressed the social disruption caused by the spread of misinformation on SNSs and its prevention. On the other hand, in some situations, the spread of corrective information fuels social disruption not the misinformation itself. As the start of the COVID-19 pandemic, misinformation in Japan spread from the end of February to the beginning of March 2020 regarding a shortage of toilet paper caused by the pandemic, and then corrective information to the misinformation was later spread. As such information spread, toilet paper temporarily disappeared from stores [16].

However since the spread of this misinformation was rather limited, this event did not reflect the backfire effect [12]. In fact, according to a survey by the Ministry of Internal Affairs and Communications [17], only 6.2% of people believed that a toilet-paper shortage actually existed. Therefore, the social disruption of buying up large quantities of toilet paper may have

actually been caused by the spread of corrective information rather than the misinformation. No research has investigated the social disruption caused by the diffusion of corrective information. Therefore, it remains unclear what amount of corrective information should be diffused to control the social disruption caused by misinformation without subsequently causing a still greater impact by the excessive diffusion of corrective information.

In this study, we model the impacts of misinformation and corrective information on toilet-paper sales for the above events related to its shortage in the above event and clarify what kind of information triggered this social disruption. Then, we apply the proposed model to identify the ideal degree of diffusion of corrective information to minimize social disruption. Finally, we propose practical measures to achieve appropriate diffusion of corrective information and evaluate their effectiveness.

## Related research

### Spreading information on SNSs

P.N. Howard et al. [5] analyzed the relationship between SNSs and the Arab Spring, showing that SNSs played an important role in political discussions, and that important conversations on SNSs preceded major revolutionary events. Frangonikolopoulos et al. [18] also showed that SNS played an important role in the Arab Spring. The SNS, rather than public institutions or political parties, facilitated mobilization and change, which allowed activists and protesters to cooperate.

Another example of how the spread of information on SNSs has impacted society is the Ice Bucket Challenge. According to Sohn et al. [6], Ice Bucket Challenge videos were distributed on SNSs and were viewed by 440 million people worldwide, raising $220 million from 28 million people. The money positively impacted ALS research. Sohn et al. [6] also observed that one of the reasons the Ice Bucket Challenge was so successful is that it has cleverly harnessed the power of social media to quickly create new norms of socially acceptable behavior.

As shown by the above examples, the diffusion of information on SNSs affects the world in various situations. Research is investigating the diffusion of information on SNS, including what kind is most likely to be spread [19], how information is spread [20], and who plays the most important role in its diffusion [21].

### Spreading misinformation

Since much misinformation is spread on SNSs and because concern is growing about its effects, many studies have minutely analyzed its spread on SNSs. H. Allcott et al. [8] analyzed fake news during the 2016 U.S. presidential election and concluded that the average American adult may have seen one or more fake news stories in the months before the election. Soroush Vosoughi et al. [22] analyzed on Twitter the spread of correct information, partially incorrect information, and fake news on Twitter. Their results found that fake news spread faster and more widely than other types of information. Moreover, we found that political fake news spread more widely than other categories of fake news. Karishma Sharma et al. [23] analyzed the spread of COVID-19-related misinformation and showed that it spread across national borders by social networking sites. Many studies have tackled the spread of COVID-19 misinformation [24–26].

There have also been many studies on the spread of misinformation by bots. In particular, many bots can be found on Facebook and Twitter [27, 28]. Chengcheng Shao et al. [13] analyzed data obtained from Twitter during and after the 2016 U.S. presidential election and concluded that bots played an important role in the spread of fake news.

## Preventing the spread of misinformation

As described above, much misinformation is spread on SNS, and many prevention methods have been studied to prevent this. The spread of misinformation can be prevented in two main ways: correcting the perceptions of those who believe the misinformation by disseminating corrective information, and detecting its spread.

B. Nyhan et al. [29] analyzed the effect of corrective information on resolving misperceptions and changing views to become positive toward vaccinations. This work was aimed at combatting the misperception that the influenza vaccine itself causes influenza. Their results showed that although the misperceptions were eliminated, those who were originally reluctant to be vaccinated became more adamant. This result resembles the results that have been shown for the MMR vaccine against COVID-19 [30]. Much research into these issues is also ongoing outside of the vaccine arena. B. Nyhan et al. [12] analyzed the effect of corrective information on political misperceptions and showed that in many cases, the corrective information failed to undo the misperceptions. They also gave examples of backfire effects.

Research has also investigated the differences in the effects of different fields of misinformation and the nature of the corrective information. N. Walter et al. [31] showed that political and marketing misinformation are more difficult to correct than misinformation about health. They also concluded that more detailed corrective information is more effective at disrupting misperceptions.

Various features can be used to detect misinformation. According to Kai Shu et al. [32], features can be derived from either the news content or from social contexts. Those based on the former can be divided into linguistic and visual features. Linguistic features include words and sentences, and Martin Potthast et al. [33] attempted to use them to distinguish between hyperpartisan news and fake news. Other studies using such features include work by Sadia Afroz et al. [34, 35].

Visual features include photographs and videos. Aditi Gupta et al. [36] used these features to distinguish between fake and real images of Hurricane Sandy with 97% accuracy. Other studies using such features include work by Zhiwei Jin et al. [37].

Social context-based features include user features (e.g., age and number of tweets) [38], posting features (e.g., reactions to misinformation and opinions) [39], and network features (e.g., follow-follower relationships) [40]. Many studies have been conducted on each type of feature. The detection of misinformation related to the COVID-19 pandemic using the characteristics of users and posts was also studied by Al-Rakhami et al. [41].

In addition, as mentioned above, since bots are often involved in the spread of misinformation, much research has been done on their detection. For bot detection, methods using supervised learning generally give good results [14, 42]. As an unsupervised learning approach, Nikan Chavoshi et al. [15]. proposed a dynamic time warping method that detects bots based on the correlations among accounts. Their results outperformed supervised learning.

## Materials and methods

### Twitter data

In this study, we analyze the spread on Twitter of misinformation about a toilet paper shortage in Japan and its correction during the initial outbreak of the COVID-19 pandemic. Twitter is a microblogging system that allows users to send and receive short posts called "tweets". Twitter users follow other users, and when you follow someone, their tweets will appear in your Twitter "timeline". You can create your own tweets or retweet information that others have tweeted. By retweeting, you can share information with more people quickly and efficiently.

We chose Twitter because it has millions of users and name recognition in Japan and can easily spread information through various features, including retweeting. Therefore, this SNS is suitable for studying the diffusion of information. Another reason for using Twitter is that it is the social medium with the highest rate of use for information gathering in Japan, according to a survey by the Ministry of Internal Affairs and Communications (MIC) [17].

This study used two main types of Twitter data: tweet data about toilet paper shortages and data on the accounts related to these tweets and their relationships.

**Tweet data.**   These data were obtained from tweets that were likely related to the misinformation on the toilet paper shortage that occurred during COVID-19. The data were collected from February 21 to March 10, 2020. We used SearchAPI to collect tweets in Japanese that contain words related to products in short supply, such as toilet paper and tissues. We retrieved 4,476,754 tweets during the collection period, 2,945,782 of which were retweets (RTs). We used the top 10,000 RTs of the collected tweets for the analysis. In this way, we excluded from analysis those tweets with very few RTs, which are rarely seen by people, because those tweets should have little impact on purchasing behavior.

We manually classified the obtained tweets into the following five categories below based on their contents. Here, 1,868 of the top 10,000 tweets were categorized. Tweets that were not categorized included jokes and various humorous comments about the toilet paper shortage.

We chose the following five categories to clarify the relationship among misinformation, corrective information, and purchasing behavior. We included the sold-out information to investigate the effect of the factual information on purchasing behavior, which created the situation where this product, toilet paper, became unavailable in stores.

- Tweets that encourage misinformation and toilet paper hoarding (misinformation tweets)

- Tweets refuting misinformation and toilet paper shortages (corrective tweets)

- Tweets that indicate that toilet paper is actually sold out (sold-out tweets)

- Tweets that include both misinformation and incentives to buy up toilet paper, as well as content that indicates that toilet paper is actually sold out (misinformation/sold-out tweets)

- Tweets that include both denials of misinformation and toilet paper shortages, as well as content that indicates that toilet paper is actually sold out (corrective/sold-out tweets)

A summary of the data for each category is shown in Table 1. Results of the number of tweets and accounts that have retweeted (RT accounts) show that the number of tweets having correction of the misinformation (corrective tweets and corrective/sold-out tweets), is much larger than that of tweets having the misinformation (misinformation tweets and misinformation/sold-out tweets). This indicates that the information that corrected the misinformation was spread much more on Twitter than the misinformation itself.

**Follower data.**   The number of retweets is commonly used as an indicator of the spread of false information and corrections. However, the degree of spread differs depending on the

**Table 1. Tweet data summary.**

| Category | Number of tweets (excluding duplicates) | Number of RT accounts |
|---|---|---|
| Corrective tweets | 949 | 356,944 |
| Misinformation tweets | 11 | 582 |
| Sold-out tweets | 650 | 72,597 |
| Corrective/sold-out tweets | 255 | 119,141 |
| Misinformation/sold-out tweets | 3 | 207 |

number of followers of the retweeting account. We used the follower data of retweeted accounts to estimate the number of people who may have been influenced. We collected the followers of accounts that tweeted or retweeted about toilet paper shortages to estimate the number of people who may have seen the misinformation or corrective tweets.

We obtained the followers of 2,044,204 users from the Twitter API for March 13–18, 2020. Consequently, 97,430,525 accounts were collected. However, it should be noted that information about private accounts is unavailable.

Here, we assume that the number of followers of an account that tweets or retweets the target information is roughly proportional to the number of users who had a chance to see the information.

## Toilet paper sales index

The toilet paper sales index was used as an indicator of how much toilet paper was sold. This data was provided by NOWCAST, INC. The sales index of toilet paper is an index that expresses the degree of change in sales of toilet paper compared to the previous year, and it is expressed in Eq 1:

$$S = (Sales_t / Sales_{t-dt}) - 1 \tag{1}$$

where S is the sales index and $Sales_t$ is sales of toilet paper on a given day t, dt = 364 days. This index is created from point-of-sale (POS) data from 1,200 supermarkets across the country. The change in the sales index during the data analysis period is shown in Fig 1. Throughout this period, sales of toilet paper were higher than usual, with a peak on February 28.

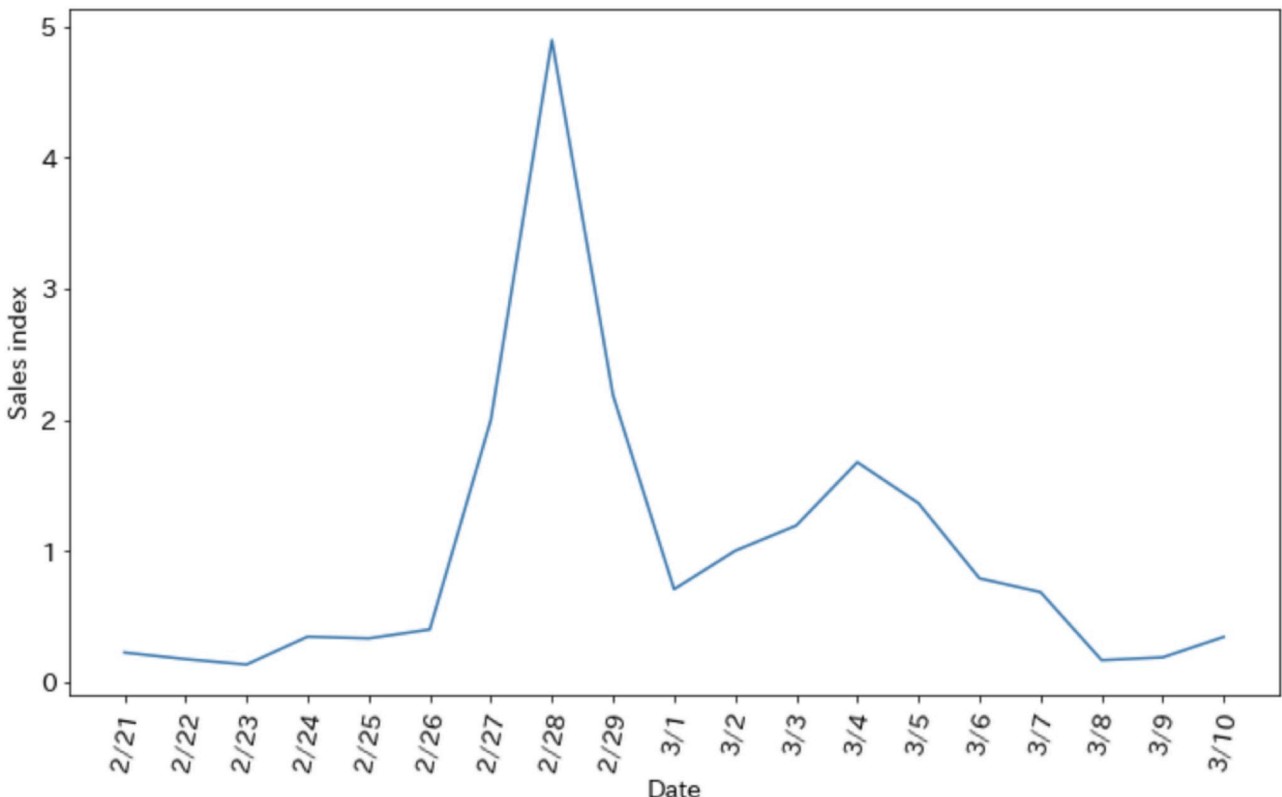

**Fig 1. Changes in sales index.**

### Flow overview

In this study, there are two phases: modeling of the relationship between information diffusion and toilet paper sales and simulation using the model. The modeling phase is referred to as Study 1, and the simulation phase is referred to as Study 2. First, in Study 1, we estimated the number of users who viewed the misinformation tweets, corrective tweets, etc. Then, we modeled the relationship between the estimated number of users and the sales index using principal component regression. In Study 2, we used a model to simulate varying the diffusion of misinformation tweets and corrective tweets, and we identified the amount of diffusion of corrective information that minimizes the impact on society. Finally, we proposed a realistic strategy that can achieve an appropriate amount of corrective information diffusion and showed its effectiveness.

## Study 1: Relationship between information diffusion and sales

### Regression model

**Estimated number of views of a tweet.** To determine how deeply tweets about the toilet paper shortage spread, we categorized the users who came into contact with the information into the following seven types:

- $x_1$: Estimated number of views of only corrective information

- $x_2$: Estimated number of views of only misinformation

- $x_3$: Estimated number of views of only sold-out information

- $x_4$: Estimated number of views of corrective information and misinformation

- $x_5$: Estimated number of views of corrective information and sold-out information

- $x_6$: Estimated number of views of misinformation and sold-out information

- $x_7$: Estimated number of views of corrective information, misinformation, and sold-out information

We obtained the number of views of each type on a daily basis. Users who are likely to view each tweet are considered the sum of the followers of a user who tweeted or retweeted, excluding duplicates. However, since not every follower actually viewed the tweets at this time, we assume that among the followers who could have viewed the tweets, those who used Twitter daily are more likely to have seen the tweets. That is, if the number of potential followers is $|U_f|$ and the daily active user rate is $r_a$, then

$$|U_t| = |U_f| \cdot r_a \tag{2}$$

The estimated number of views ($|U_t|$) is calculated by this formula. For active user rate $r_a$, we used 24%, which is the value of the Monetizable Daily Active Usage (mDAU) (published by Twitter every quarter) for the first quarter of 2020.

Fig 2 shows an image of the users who viewed tweets. Assume a user who posted a misinformation tweet and a corrective tweet. The misinformation tweet was retweeted by one user, and the corrective tweet was retweeted by one user. We assume that each retweeted user has two followers. However, the user who retweeted the misinformation and one of the users who retweeted the corrective tweet share a follower. For the situation shown in this figure, the following is the estimated number of views:

- Estimated number of views of only corrective information $x_1 = 3 \cdot r_a$

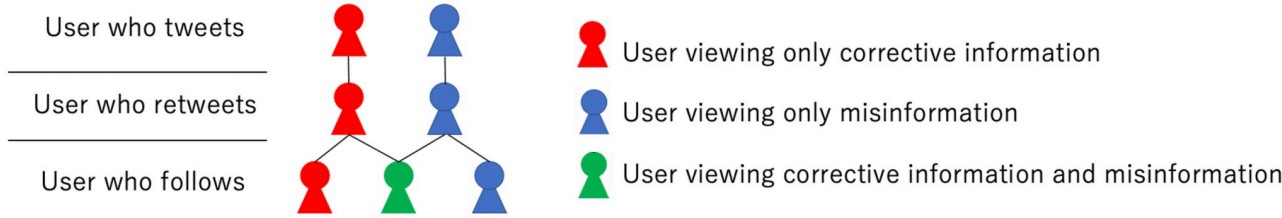

**Fig 2. Illustration of the estimated number of users viewing tweets.**

- Estimated number of views of only misinformation information $x_2 = 3 \cdot r_a$
- Estimated number of views of corrective information and misinformation $x_4 = 1 \cdot r_a$

Fig 3 shows the estimated number of views during the coverage period. The peaks of the estimated numbers of views of only corrective information $x_1$ and the corrective and sold-out information $x_5$ are on February 28, which coincides with the peak of the sales index. In addition, the estimated number of views of only corrective information ($x1$), the estimated number of views of only sold-out information ($x_3$), and the estimated number of views of the corrective and sold-out information ($x_5$) were almost non-existent throughout the period.

Table 2 shows the total estimated number of views. We round figures down to the nearest whole number. The total estimated number of views of the corrective and sold-out information is the largest, followed by the total estimated number of views of only the corrective

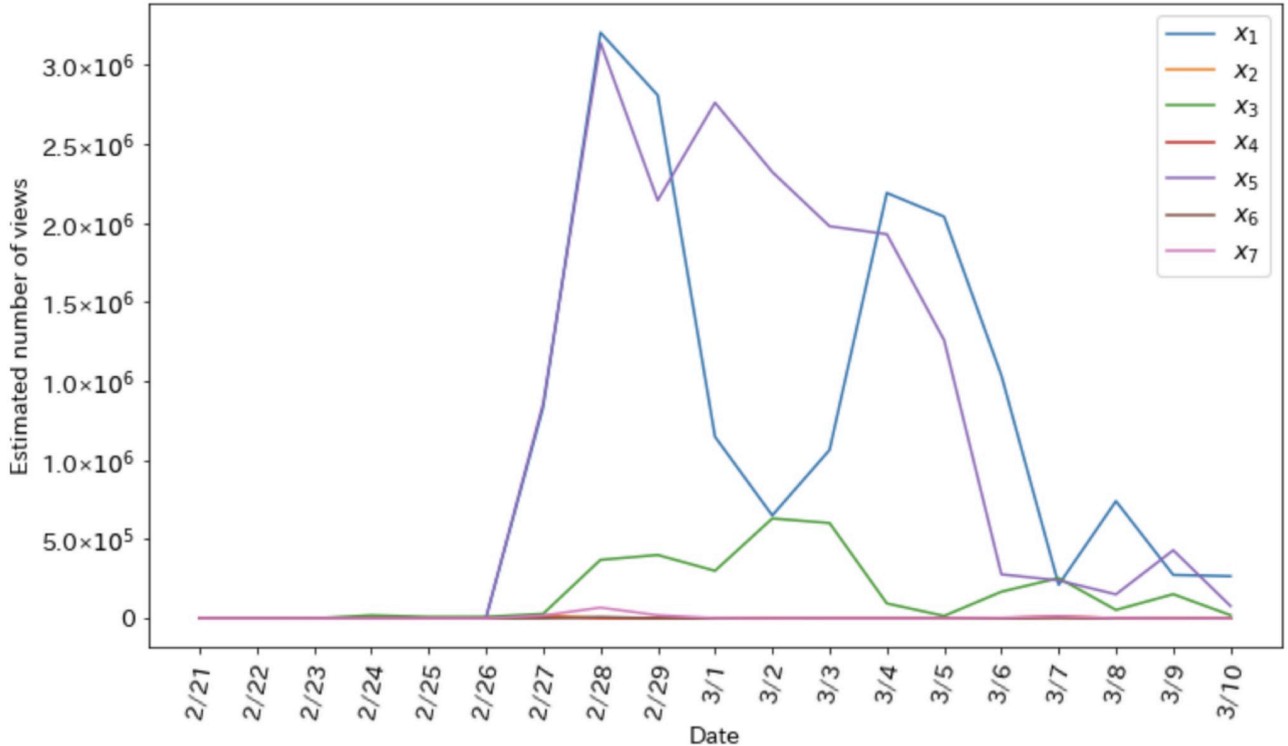

**Fig 3. Estimated number of views of each type.**

**Table 2. Total estimated number of views of each type.**

| | Total estimated number of views |
|---|---|
| Only corrective information | 19,461,241 |
| Only misinformation | 41,988 |
| Only sold-out information | 3,133,407 |
| Corrective information and misinformation | 17,312 |
| Corrective information and sold-out information | 21,559,552 |
| Misinformation and sold-out information | 17,953 |
| Corrective information, misinformation and sold-out information | 128,889 |

information. The total estimated number of views of only the sold-out information is also found at a certain number. The total estimated number of views of the corrective information and misinformation was the smallest, followed by the total estimated number of views of the estimated and sold-out information.

Note that, in the total number of tweets, if a user sees only the corrective tweets on one day and the sold-out tweets on the next day, she is counted as both total estimated number of views of only the corrective information and total estimated number of views of only the sold-out information.

**Principal component analysis.** Based on the change in the estimated number of views, we modeled the relationship among the spread of misinformation, sold-out, and corrective information, and toilet-paper sales to reveal what kind of information contributed to excessive sales.

In this section, we built a model that explains the daily toilet-paper sales index in terms of the number of estimated views for the day. However, as shown in Table 3, the correlation among the explanatory variables was too large. Therefore, to avoid multicollinearity, we conducted principal component analysis and regression analysis using the obtained principal components as explanatory variables.

Table 4 shows the eigenvectors for each principal component, and Table 5 shows the contribution rate of each one. The components of the eigenvectors show the correlation between each principal component and $x_1$ to $x_7$. The contribution rate of the first principal component is as large as $8.75 \times 10^{-1}$, indicating that it can explain most of the original data. However, to analyze the impact of diffusion information on sales, only modeling the first principal component is inappropriate, so the principal components necessary for explanation must be employed in the model. Using the backward stepwise selection, the set of principal components with p-values less than 0.05 was used as explanatory variables. We obtained the first, second, and fourth principal components.

**Table 3. Correlation between estimated views.**

| | $x_1$ | $x_2$ | $x_3$ | $x_4$ | $x_5$ | $x_6$ | $x_7$ |
|---|---|---|---|---|---|---|---|
| $x_1$ | 1.00 | 0.04 | 0.36 | 0.22 | 0.76 | 0.39 | 0.64 |
| $x_2$ | 0.04 | 1.00 | -0.03 | 0.95 | -0.02 | 0.27 | 0.33 |
| $x_3$ | 0.36 | -0.03 | 1.00 | 0.01 | 0.73 | 0.24 | 0.30 |
| $x_4$ | 0.21 | 0.95 | 0.01 | 1.00 | 0.08 | 0.33 | 0.48 |
| $x_5$ | 0.76 | -0.02 | 0.73 | 0.08 | 1.00 | 0.30 | 0.49 |
| $x_6$ | 0.39 | 0.27 | 0.24 | 0.33 | 0.30 | 1.00 | 0.87 |
| $x_7$ | 0.64 | 0.33 | 0.30 | 0.48 | 0.49 | 0.87 | 1.00 |

**Table 4. Eigenvectors of principal components.**

|  | $x_1$ | $x_2$ | $x_3$ | $x_4$ | $x_5$ | $x_6$ | $x_7$ |
|---|---|---|---|---|---|---|---|
| 1st principal component | 0.66 | 0.00 | 0.07 | 0.00 | 0.74 | 0.00 | 0.01 |
| 2nd principal component | -0.75 | 0.00 | 0.17 | 0.00 | 0.64 | 0.00 | -0.01 |
| 3rd principal component | -0.08 | 0.00 | -0.98 | 0.00 | 0.17 | 0.00 | -0.01 |
| 4th principal component | 0.01 | 0.18 | -0.01 | 0.07 | 0.00 | 0.14 | 0.97 |
| 5th principal component | 0.00 | 0.93 | 0.00 | 0.30 | 0.00 | -0.07 | -0.19 |
| 6th principal component | 0.00 | 0.11 | 0.00 | -0.22 | 0.00 | 0.96 | -0.14 |
| 7th principal component | 0.00 | 0.29 | 0.00 | -0.93 | 0.00 | -0.24 | 0.05 |

The eigenvector of the first principal component is positively correlated with $x_1$ and $x_5$, indicating an increase or decrease in the estimated number of views of only the corrective information and the corrective and sold-out information.

The eigenvector of the second principal component is negatively correlated with $x_1$ and positively correlated with $x_5$, indicating the difference between the increase and decrease in the estimated number of views of corrective and sold-out information, and the number of views of only corrective information. The eigenvector is weakly positively correlated with $x_3$, indicating that it also takes into account the increase or decrease in the estimated views of only the sold-out information.

The eigenvector of the fourth principal component is positively correlated with $x_2$, $x_6$, and $x_7$, indicating an increase or decrease in the estimated number of views of only the misinformation, the misinformation and sold-out information, and the corrective information, misinformation, and sold-out information.

**Importance of variables.** In ordinary multiple regression, it is common to measure importance by the magnitude of the standardized partial regression coefficient. However, in this study, since the principal component obtained through principal component analysis was used as an explanatory variable, it is not available. Therefore, we evaluated the importance by using the inner product of the standardized partial regression coefficients and the eigenvectors [43]. If there are $I$ principal components, $\beta_j$ denotes the importance of each estimated number of views, and $c_{ij}$ denotes the j component of the eigenvector of the i principal component. $\alpha_i$ is the standardized partial regression coefficient:

$$\beta_j = \sum_{i=1}^{I} c_{ij}\alpha_i \tag{3}$$

There are two interpretations of importance: one is the impact of toilet paper sales on the increase or decrease in the number of users exposed to each piece of information, and the

**Table 5. Contribution rate of each principal component.**

|  | Contribution rate |
|---|---|
| 1st principal component | $8.75 \times 10^{-1}$ |
| 2nd principal component | $1.20 \times 10^{-1}$ |
| 3rd principal component | $4.79 \times 10^{-3}$ |
| 4th principal component | $5.03 \times 10^{-5}$ |
| 5th principal component | $7.08 \times 10^{-6}$ |
| 6th principal component | $2.97 \times 10^{-7}$ |
| 7th principal component | $2.30 \times 10^{-8}$ |

**Table 6. Regression coefficients, p-values, and t-values.**

|  | $a_1$ | $a_2$ | $a_3$ | b |
|---|---|---|---|---|
| Coefficients | $1.36 \times 10^{-7}$ | $-9.04 \times 10^{-7}$ | $1.00 \times 10^{-7}$ | 0.9919 |
| P-values | 0.000 | 0.000 | 0.001 | 0.000 |
| T-values | 16.76 | -4.14 | 9.40 | 17.88 |

other is the impact of the increase or decrease in the number of users exposed to each piece of information on toilet paper sales. Taking the example of corrective information, it is more convincing to assume that people will make purchases even if the corrective information spreads, rather than that the corrective information spreads as purchases increase. This is because the idea that information about the existence of a misinformation would have caused people to anticipate a shortage of toilet paper and thus accelerate their purchases seems more likely than the opposite. Therefore, we consider the latter interpretation to be more convincing and likely, and assume that the importance is the magnitude of the impact of the increase or decrease in the number of users exposed to each piece of information on the toilet paper sales.

## Results and discussion

Table 6 shows the coefficients, the p-values, and the t-values obtained from the principal component regression. $a_1$, $a_2$, and $a_3$ respectively denote the regression coefficients of the first, second, and fourth principal components, and $b$ denotes the constant term. The p-values for all coefficients are less than 0.05.

Fig 4 shows the regression results and Table 7 shows the coefficient of determination and the F-value of the model. The regression is highly accurate.

Table 8 shows the importance of each variable. The estimated number of views of only the corrective information ($x_1$) is clearly the most important, followed by the estimated views of both the corrective and sold-out information ($x_5$). In other words, an increase in the number of users who saw the corrective information or the corrective and sold-out information accelerated the purchase of toilet paper, which would likely lead to a shortage.

Here the strongest influence comes from users who saw only the corrective information, which means that the influence of those who only came into contact with the information that misinformation about toilet paper shortages was spreading, even though in fact there was no shortage is extremely strong. Although there is a backfire effect [12] in which beliefs are strengthened by receiving corrective information about misinformation, the fact that the strongest impact came from users who were not exposed to the original misinformation suggests that in this case, the backfire phenomenon was not the cause of the social disruption.

So far, we have shown that the mass diffusion of corrective information itself caused the social disruption of toilet-paper hoarding. One factor that may have caused this phenomenon is pluralistic ignorance [44]. Pluralistic ignorance denotes a situation in which the majority of a group denies certain arbitrary conditions, while also assuming that others accept them and act accordingly. In this event, the majority of people did not believe the misinformation about toilet paper. However, they thought that others might believe the misinformation, which led them to anticipate that toilet paper might be sold out, prompting them to purchase it for themselves.

In the next section, we discuss a method that estimates the optimal diffusion of corrective information, which can spark social disruption.

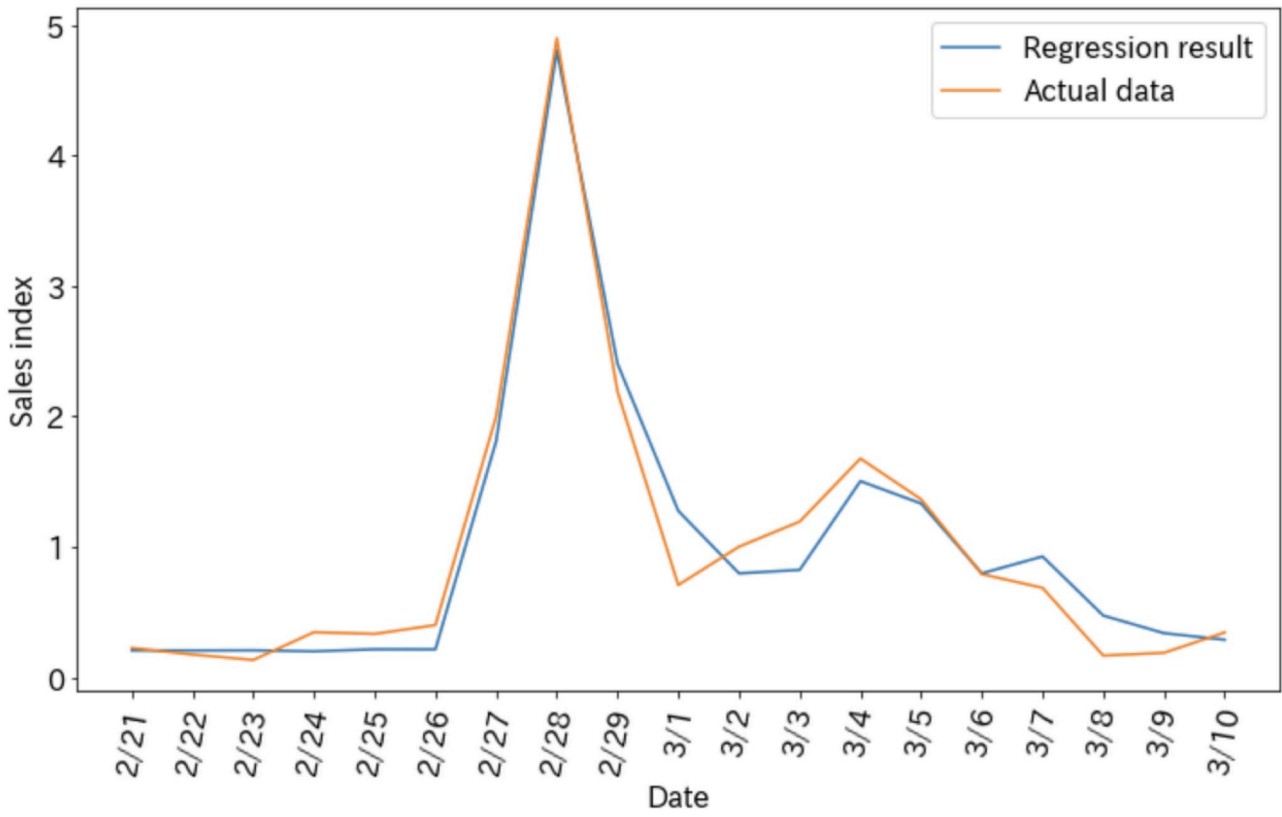

**Fig 4. Regression results.**

## Study 2: Estimating the optimal diffusion rate

### Experiment on reducing the number of users who spread corrective information

**Aim of the experiment.** The previous section described the danger of corrective information that causes social disruption was described. In this section, using the model from the previous section, we discuss policies that reduce social disruptions.

**Table 7. Coefficient of determination and F value.**

| Coefficient of determination | F value |
|---|---|
| 0.963 | 128.9 |

**Table 8. Importance of variables.**

|  | Importance |
|---|---|
| $x_1$ | $70.10 \times 10^{-2}$ |
| $x_2$ | $8.66 \times 10^{-2}$ |
| $x_3$ | $2.11 \times 10^{-2}$ |
| $x_4$ | $3.28 \times 10^{-2}$ |
| $x_5$ | $49.50 \times 10^{-2}$ |
| $x_6$ | $6.70 \times 10^{-2}$ |
| $x_7$ | $46.10 \times 10^{-2}$ |

First, we used the proposed model to simulate the impact of the changes in the diffusion of the corrective tweets as well as the corrective/sold-out tweets (hereinafter, correction-related tweets) on the sales index.

**Methods.** In this simulation, we checked how the sales index changed when the number of users who retweeted corrective tweets changed.

First, we built a multi-agent model where one user corresponds to one agent. In this case, each agent has a follower-relation network based on actual data. When each agent comes in contact with various types of information (the original tweet), she chooses whether to spread (retweet) it to other agents to whom she is connected on the network. Her neighboring agents who can spread the information decide whether to do so with a certain probability as well. Since each agent corresponds to an actual user, it is possible to obtain the same result as the actual data by deterministically deciding whether the agent actually diffused the information.

In the constructed agent model, we changed the probability that each agent retweets a correction-related tweet and calculated the sales index using the estimated number of each view type, $x_1 - x_7$, obtained by the change.

In the simulation, we used the follower relationship and who retweeted from actual data. However, some agents do not retweet correction-related tweets with a certain probability, regardless of their behavior in the actual data. The percentage of agents who RT from the actual data is the RT rate. In this experiment, RT rates were set at 75%, 50%, 25%, and 0%. In the case of an RT rate of 50%, the diffusion is calculated when half of the users who could retweet in reality do not retweet.

**Results and discussion.** Fig 5 shows the daily changes in the sales index when the RT rate is varied. The smaller the RT rate is, the smaller is the estimated sales index, which is minimized at the RT rate of 0%.

However, from a practical point of view, if the number of correction-related tweets decreases, the number of views of only the misinformation will increase, and the spread of misinformation tweets and misinformation/sold-out tweets (hereinafter misinformation-related tweets) will also increase. In the next section, we experimentally investigate this situation.

## Optimal diffusion rate estimation experiment

**Aim of experiment.** The excessive diffusion of corrective information may increase social disruption. The previous experiment showed that if the RT rate is 0%, i.e., all users who could actually RT do not, the sales index is minimized.

However, it remains unclear to what extent it is generally desirable to spread corrective information. If the misinformation is too ridiculous or trifling, the diffusion rate is low and the effect is small; if the misinformation seems highly credible or appears important, the diffusion rate is high and the risk increases. Accordingly, the optimal diffusion rate of corrective information is expected to vary depending on the diffusion rate of the misinformation. Therefore, we performed a simulation to estimate the optimal diffusion rate of the corrective information based on the diffusion rate of misinformation.

Then we conducted a simulation that investigated the change in the sales index by varying the diffusion of the correction- and misinformation-related tweets. We changed the diffusion by altering the users who retweeted the correction- and misinformation-related tweets.

**Methods.** As in the experiment on reducing the number of users who spread corrective information, we constructed a multi-agent model where one user corresponds to one agent. In this case, each agent has a follower-relationship network of agents based on actual data. Among the agents, only those who contacted the source of various information (the original tweet) chose whether to spread (retweet) the information.

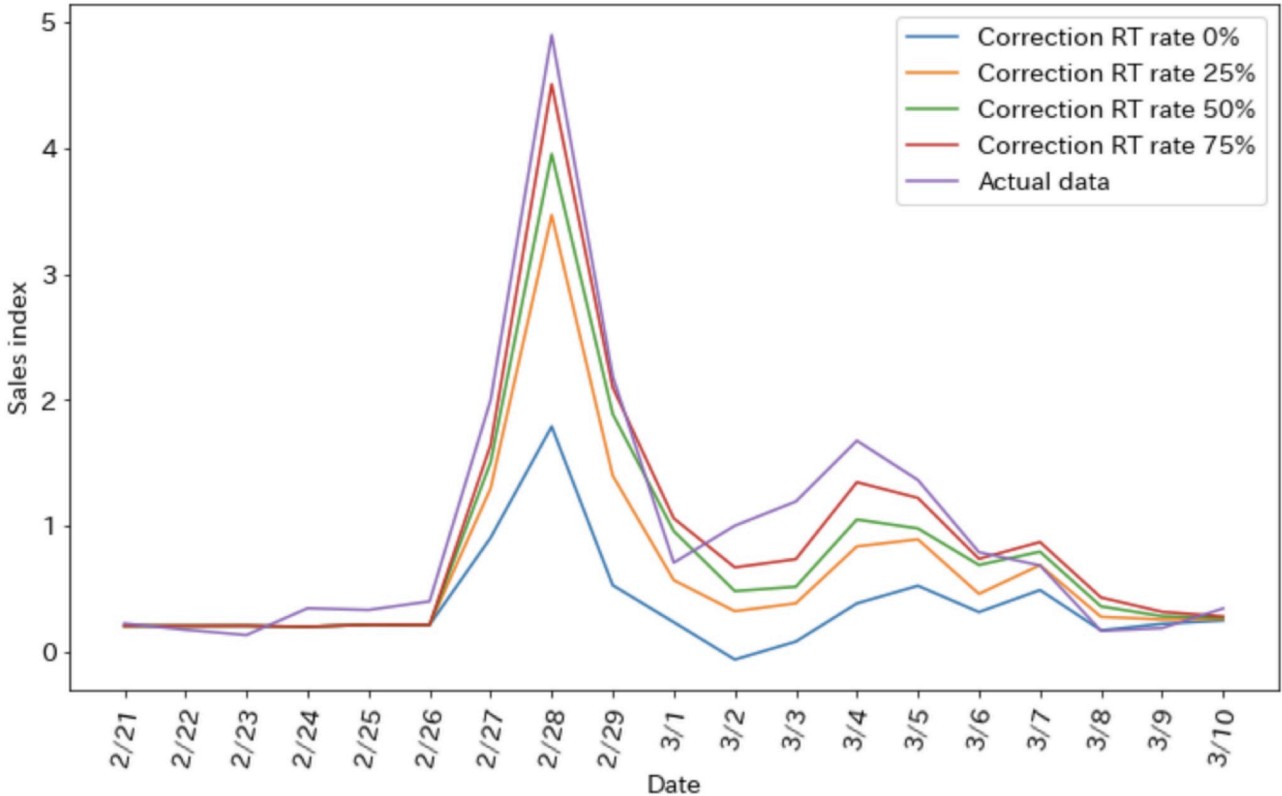

**Fig 5. Results of experiment.**

In the agent model we constructed, we increased or decreased the probability of retweeting both correction- and misinformation-related tweets and calculated the sales index using the estimated number of each view type, $x_1 - x_7$. In addition, the agents who saw the correction-related tweets before the misinformation-related tweets did not spread the misinformation tweets.

We call the parameters of the correction-related tweets and the misinformation-related tweets the correction RT rate and the misinformation RT rate. In this experiment, the correction RT rates were 0%, 0.4%, 0.8%, 1.2%, 1.6%, and 2.0%, and the misinformation RT rates were 0%, 1%, 5%, 10%, and 15%.

With these combinations, experiments were conducted under 30 different conditions. Ten were conducted for each experimental condition, and the mean of the sum of the estimated sales indices during the experimental period was used as the result.

**Results and discussion.** Fig 6 shows the analysis results of the impact of misinformation on sales. The vertical axis shows the sum of the sales indices during the period. When the misinformation RT rate is small (0.0%, 1.0%, or 5.0%), the sales index increases as the correction RT rate increases. On the other hand, when the misinformation RT rate is large ($\geq 10$%), the sales index decreases at the beginning and increases in the middle as the correction RT rate increases. In other words, when the credibility of the misinformation is low, the spread of corrective information should be suppressed as much as possible, but not when the credibility of the misinformation is high. When the misinformation's credibility is high and there is no corrective information, the sales index increases due to the influence of the misinformation. Therefore, the diffusion of a certain amount of corrective information is necessary.

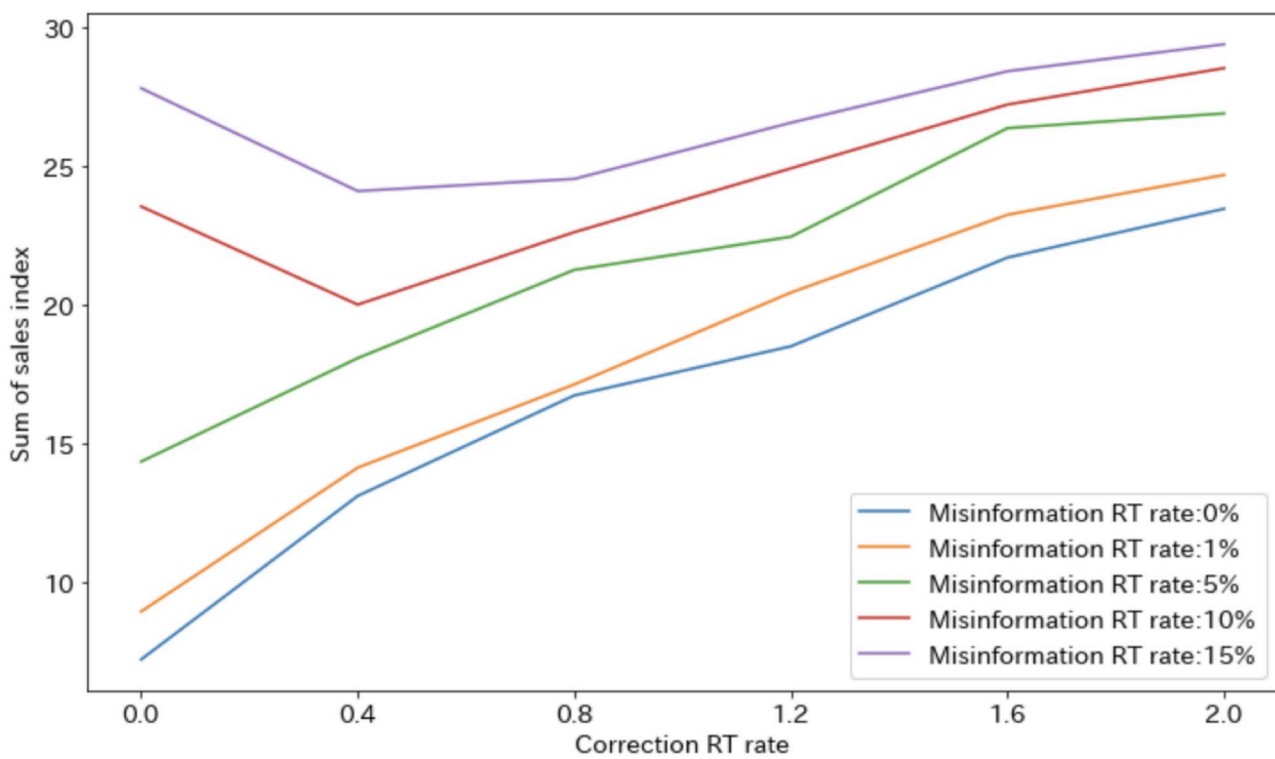

**Fig 6. Results of optimal diffusion rate estimation experiment.**

Fig 7 shows the relationship between the correction RT rate and the estimated number of views of the correction-related tweets for each misinformation rate. The estimated views of the correction-related tweets is hardly affected by the misinformation tweet rate. Therefore, regardless of the misinformation RT rate, if correction-related tweets spread, the number of users who are exposed to the corrective information will increase accordingly.

On the other hand, Fig 8 shows the relationship between the correction RT rate and the estimated number of views of the misinformation-related tweets for each misinformation RT rate. When the misinformation RT rate is high ($\geq$10%), the number of views of misinformation-related tweets decreases as the correction RT rate increases. However, an increase in the correction RT rate above a certain level does not increase the misinformation contact agents. Although retweeting correction-related tweets does suppress misinformation, not every trace of such misinformation can be contained. This conclusion can be understood by considering the fact that a certain number of agents first come into contact with the misinformation rather than the corrective information.

On the other hand, when the misinformation RT rate is small (0.0%, 1.0%,or 5.0%), there is no significant change in the estimated number of views of the misinformation-related tweets regardless of the correction RT rate. This is because the number of users who originally retweet misinformation tweets is small, and thus the effect of the diffusion of corrective-related tweets to suppress the diffusion of misinformation is small.

From Figs 7 and 8, when the misinformation RT rate is high, if the correction RT rate is small, the misinformation will increase the sales index; if the correction RT rate is high, the corrective information will increase the sales index. The result is a downward convex change in the sales index (Fig 6). In addition, when the misinformation RT rate is small, the sales

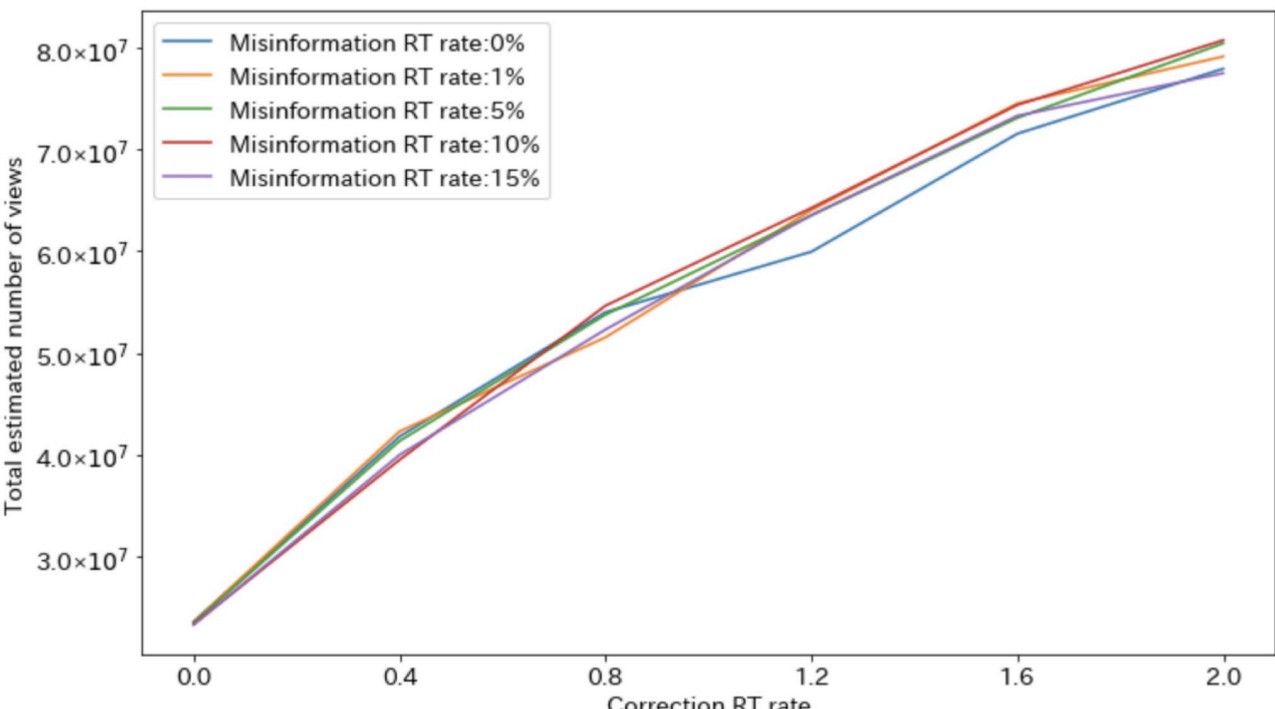

**Fig 7. Total estimated number of views of correction-related tweets.**

index increases due to corrective information regardless of the correction RT rate, leading to an increase in the sales index (Fig 6).

As shown in Fig 8, an increase in the number of correction-related RTs can reduce the spread of misinformation. However, as shown in Fig 6, the corrective information does not necessarily lead directly to social disruption, i.e., a decrease in sales. In other words, social disruption cannot be suppressed unless an appropriate amount of corrective information is spread based on the misinformation RT rate, i.e., the credibility of the misinformation.

When misinformation is spread, we must estimate the impacts of the misinformation and the corrective information on society and then carefully consider how to respond.

### Strategy to reduce spread of information and its impact

**Aim of the experiment.** The previous section showed that there is an appropriate amount of diffusion for corrective information. However, controlling the amount of diffusion is complicated in social media, where each individual user can freely spread information.

Therefore, we are considering guidelines to ensure that the impact on society is appropriate, if not optimal.

**Methods.** We assume that the excessive proliferation of correction-related tweets is caused by unrelated, well-meaning users. In other words, we assume that users who have not come into contact with the misinformation will also spread the corrective information unprincipledly, and that users who do not know the existence of the misinformation will also believe that the misinformation exists and act accordingly. Furthermore, we consider a spreading method that does not affect users who are unrelated to the misinformation. In this section, we propose a guideline where only users who saw the misinformation-related tweet (before they retweeted

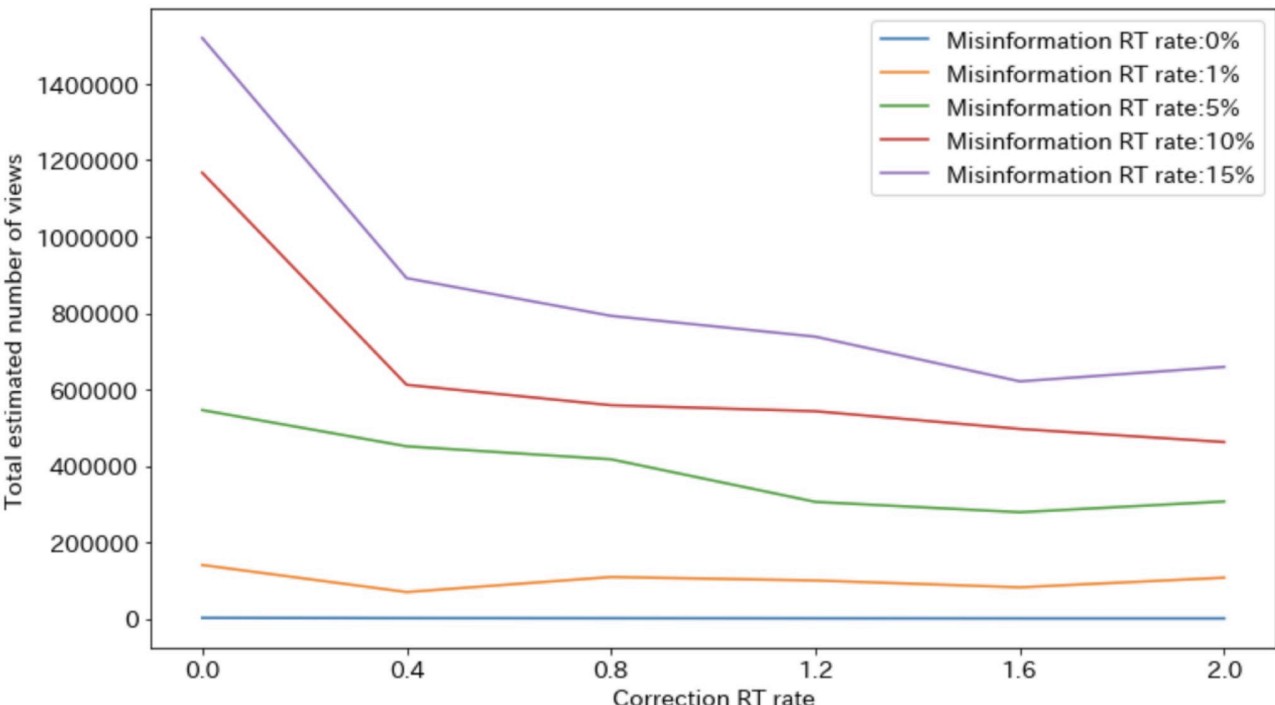

**Fig 8. Total estimated number of views of misinformation-related tweets.**

the correction-related tweet) are allowed to retweet the correction-related tweet and clarify how this changes the sales index.

As before, we built a multi-agent model where a single user corresponds to a single agent. In this case, each agent has a follower-relationship network of agents based on actual data.

In the agent model we constructed, only an agent corresponding to a user who saw the misinformation-related tweet before retweeting the correction-related tweet in the actual data is allowed to retweet the correction-related tweet. In addition, we assume that only the agent corresponding to a user who retweeted the misinformation-related tweet in the actual data retweets the misinformation-related tweet. The resulting number of each user view type, $x_1 - x_7$, is used to calculate the sales index.

**Results and discussion.** We compare the sales index estimated from the simulation and the sales index of the actual data in Table 9. The results show that the RT behavior following our proposed guidelines reduced the sales index to 48.7% of the actual data. Table 10 shows the estimated number of views of the correction- and misinformation-related tweets in the experiment and the actual data. We round down those values to the nearest whole number. The proposed guideline significantly reduced the estimated number of views of the correction-related tweets but increased the estimated views of the misinformation tweets. This is due to the fact that the diffusion of correction-related tweets decreased in proportion to the diffusion of misinformation-related tweets. This result indicates that the proposed guideline is useful.

**Table 9. Comparison of sales index.**

| Acutual data | Proposed guideline | Ratio |
|---|---|---|
| 18.85 | 9.18 | 48.7% |

**Table 10. Comparison of the estimated number of views.**

| | Actual data | Proposed guideline | Amount of change |
|---|---|---|---|
| Estimated number of views of correction-related tweets | 41,020,794 | 12,775,262 | -28,245,532 |
| Estimated number of views of misinformation-related tweets | 59,942 | 107,372 | +47,430 |

## Discussion

In this study, we analyzed the misinformation about a toilet paper shortage in Japan that occurred during the 2020 COVID-19 pandemic using tweet and sales data in order to clarify the impact of misinformation and its corrective information on society. In Study 1, we modeled the relationship between information diffusion and toilet paper sales. In the process, it became clear that the number of corrective information viewers was much larger than that of misinformation viewers. One of the contributions of this study is that it clarified that the cause of the problem was not the original misinformation but subsequent corrective information, although there were many reports in the Japanese media that the purchasing behavior for toilet paper was promoted by misinformation.

In addition, a sales forecasting model was constructed, and it was shown that the coefficient of determination was 0.963 and the F-value was 128.9 with high accuracy. Furthermore, from the coefficients of the regression equation and other data, it was found that the explanatory variables with high importance were estimated numbers of views of only corrective information and the corrective and sold-out information. This means that the corrective information, not the misinformation, promoted the purchase of toilet paper.

Mendoza et al. [45] found that a lot of corrective information was spread, but there was no analysis of the impact of that information on society. In addition, Nyhan et al. [29, 30] examined the effect of corrective information on attitudes toward hoaxes, showing that this depends on the degree to which people originally believed the misinformation. Furthermore, Nyhan et al. [12] found that providing corrective information to people who believe in misinformation can lead them to believe more strongly in the misinformation more (the backfire effect). However, in the case of the event investigated in this study, the backfire effect cannot occur because the misinformation was hardly spread in the first place. It is also possible that this event is not due to the backfire effect, but due to pluralistic ignorance [44]. Thus, no previous study has shown that corrective information had a negative impact on society, which is one of the contributions of this study.

In Study 2, we used the obtained model to estimate the sales index when the number of users who RT misinformation or corrective information changes. The results show that when the credibility of misinformation is high, corrective information has the effect of reducing over-purchases, but the excessive diffusion of corrective information promotes over-purchases. To the best of our knowledge, no study has shown that the appropriate amount of corrective information differs depending on the amount of hoax diffusion, and this is one of the contributions of this study.

Finally, as a practical strategy, we confirmed how sales index would change if we set a rule that users who do not see misinformation will not RT the corrective tweet. The results show that the sales index could be reduced to 48.7% compared to the case without the strategy. This indicates that when there are many people who do not believe in misinformation, only the users who have seen the misinformation can spread the corrected information, thereby suppressing the negative impact on society.

## Conclusion

In this study, we analyzed the toilet paper shortage misinformation in Japan that occurred during the 2020 COVID-19 pandemic using tweet and sales data to reveal the impact of the hoax and its correction information on society. As a result, we found that in this event, the corrective information promoted unnecessary purchasing behavior. In addition, we found that the appropriate amount of corrective information differs depending on the amount of false information diffused.

In the future, appropriate correction strategies must be identified based on the dissemination rate of the misinformation. In this study, we assumed a causal relationship between the diffusion of information and purchase behavior, but as a future issue, we need to investigate the causal relationship separately. In addition, over-purchasing is not the only disruption that misinformation brings to society. We must also analyze the factors that affect the social impact of various kinds of misinformation as well and identify their general tendencies to optimize corrective information. Furthermore, misinformation is not only influenced by social media but also by mass media [46]. The establishment of countermeasures against misinformation that take into account influences other than social media remains another future issue.

## Supporting information

**S1 Data.**
(CSV)

## Acknowledgments

We would like to thank NOWCAST, INC. for providing us with sales index data of toilet paper.

## Author Contributions

**Conceptualization:** Ryusuke Iizuka, Fujio Toriumi, Mao Nishiguchi.

**Data curation:** Fujio Toriumi.

**Formal analysis:** Ryusuke Iizuka.

**Methodology:** Ryusuke Iizuka, Fujio Toriumi, Mao Nishiguchi, Masanori Takano.

**Project administration:** Ryusuke Iizuka, Fujio Toriumi.

**Resources:** Mitsuo Yoshida.

**Supervision:** Fujio Toriumi, Mao Nishiguchi.

**Validation:** Ryusuke Iizuka.

**Visualization:** Ryusuke Iizuka.

**Writing – original draft:** Ryusuke Iizuka.

**Writing – review & editing:** Ryusuke Iizuka, Mao Nishiguchi, Masanori Takano, Mitsuo Yoshida.

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
