## [Decision Letter · Decision Letter 0]

1 Sep 2021

PONE-D-21-18422

Impact of Correcting Misinformation on Social Disruption

PLOS ONE

Dear Dr. Iizuka,

Thank you for submitting your manuscript to PLOS ONE. After careful consideration, we feel that it has merit but does not fully meet PLOS ONE’s publication criteria as it currently stands. Therefore, we invite you to submit a revised version of the manuscript that addresses the points raised during the review process.

The manuscript needs a MAJOR REVISION. Please follow the suggestions given by the reviewers. In particular, the manuscript should be improved by improving the description and by adding a comparison with existing methodologies.

We look forward to receiving your revised manuscript.

Kind regards,

Barbara Guidi

Academic Editor

PLOS ONE

Journal Requirements:

3. Thanks for stating that you'll provide the Tweet IDs upon accept.

Please ensure that you are complying with Twitter's Terms of Service (https://twitter.com/en/tos) in regards to data sharing.

If there are restrictions in publicly sharing Tweet IDs, please do include an updated Data Availability Statement in your cover letter.

We will update your Data Availability Statement on your behalf if needed.

Reviewers' comments:

Reviewer's Responses to Questions

**Comments to the Author**

1. Is the manuscript technically sound, and do the data support the conclusions?

Reviewer #1: Yes

Reviewer #2: Partly

Reviewer #3: Yes

2. Has the statistical analysis been performed appropriately and rigorously? 

Reviewer #1: Yes

Reviewer #2: Yes

Reviewer #3: Yes

3. Have the authors made all data underlying the findings in their manuscript fully available?

Reviewer #1: Yes

Reviewer #2: No

Reviewer #3: Yes

4. Is the manuscript presented in an intelligible fashion and written in standard English?

Reviewer #1: Yes

Reviewer #2: No

Reviewer #3: Yes

5. Review Comments to the Author

Reviewer #1: The authors models the effects of misinformation reduction and the diffusion of corrective information on social disruption and clarifies their effects on social disruption. This is a very interesting research.

The manuscript is concise and well written.

My major concerns are as follows:

(1) The authors stated that, the major contribution of the study is the "impact of corrective information on society and clarifies its appropriate amount as against the previous studies that concentrated on the impact of corrective information on attitudes toward misinformation". I think the author need to give more clarification on the bases of their justification. For example, what are the issues in the previous studies that the authors were able to identify and improve upon in this study?

(2) The authors fails to make comparism of their methods with the existing models in the research domain to demonstrate the effectiveness of their methods.

Reviewer #2: # This paper need a lot of English editing and improving the flow of the work.

# The Discussion section need to be expanded and more informative. The discussion of the main findings must be clear.

# The whole paper must be re-arranged and organized like this for example (Abstract, Introduction, Method, Results, Discussions and finally Conclusions. In the introduction the related works, the research gaps and the main objective on this work must be clearly reported. All things related to the dataset and the regression methods should be written in the methodology.

# I advised the authors to check some published examples in order to know how to arrange their work.

Reviewer #3: 1. Please reduce the Figure size and keep them clear.

2. Please check few typing mistakes such as page no. 2 line no. 61.

3. Please try to keep the tables near to the text so that it would be easy to read.

6. PLOS authors have the option to publish the peer review history of their article (what does this mean?). If published, this will include your full peer review and any attached files.

Reviewer #1: No

Reviewer #2: No

Reviewer #3: **Yes: **Krishna Mohan Kudiri

---

## [Author Response · Author response to Decision Letter 0]

9 Jan 2022

Dear Editor and Reviewers:

 Thank you for reviewing our paper, “Impact of correcting misinformation on social disruption.” We appreciate all the comments, which were constructive and helpful for improving the manuscript. Please refer to our responses below and the revised parts in the main paper. The revised parts are highlighted.

We trust that our revisions are satisfactory, but we are happy to consider further amendments if requested.

Reviewer #1

(1) The authors stated that, the major contribution of the study is the "impact of corrective information on society and clarifies its appropriate amount as against the previous studies that concentrated on the impact of corrective information on attitudes toward misinformation". I think the author need to give more clarification on the bases of their justification. For example, what are the issues in the previous studies that the authors were able to identify and improve upon in this study?

Response 1:

Thank you for your insightful comments. 

Our contributions can be briefly summarized as follows

1) We observed that people behaved as if they had believed misinformation even if the original misinformation was not spread to them. 

2) Our results suggest that such behavior is caused by pluralistic ignorance due to spreading excessive corrective information.

3) Under such a situation, we used simulation experiments to show that the appropriate amount of corrective information to spread depends on the amount of misinformation spread.

We described the differences between previous studies and our study in more detail in the Discussion section (p.15-16, ll.554-596), in order to clarify our contribution. 

(2) The authors fails to make comparism of their methods with the existing models in the research domain to demonstrate the effectiveness of their methods.

Response 2:

The main contribution of our study is not the proposal of any new methods, but the detection of new phenomena. The method used is principal component regression, which is often used and works well. Therefore, it is not necessary for us to newly verify the effectiveness of this method.

Reviewer #2: 

# This paper need a lot of English editing and improving the flow of the work.

Response 1:

Thank you for providing important comments.

As for the English, the text was revised by a professional editing service.

Regarding the flow of the work, we have revised the text to match the typical section structure as follows. 

A description of the overall flow of this research has been added to the Materials and Methods section. We wrote about the statistical analysis in Study 1 and about the simulation in Study 2. In Study 1, we described the modeling method and its results. Study 2 covers the purpose, methods, results, and discussion of each of the three simulations.

# The Discussion section need to be expanded and more informative. The discussion of the main findings must be clear.

Response 2:

Our main findings are as follows:

1) We observed that people behaved as if they had believed misinformation even if the original misinformation was not spread to them. 

2) Our results suggest that such behavior is caused by pluralistic ignorance due to spreading excessive corrective information.

3) Under such a situation, we used simulation experiments to show that the appropriate amount of corrective information to spread depends on the amount of misinformation spread.

We clarify these findings by discussing the consistencies and differences between our results and previous works (see the Discussion section (p.15-16, ll.554-596)).

# The whole paper must be re-arranged and organized like this for example (Abstract, Introduction, Method, Results, Discussions and finally Conclusions. In the introduction the related works, the research gaps and the main objective on this work must be clearly reported. All things related to the dataset and the regression methods should be written in the methodology.

# I advised the authors to check some published examples in order to know how to arrange their work.

Response 3:

According to this comment, we revised the section structure of our manuscript based on the conventions in the literature (see Response 1).

Reviewer #3: 

1. Please reduce the Figure size and keep them clear.

Response 1:

Thank you for your helpful comments.

The size of the figures may be due to the submission system of PLOS ONE. When published, they will be of the appropriate size.

2. Please check few typing mistakes such as page no. 2 line no. 61.

Response 2:

We checked this and made the necessary revisions.

3. Please try to keep the tables near to the text so that it would be easy to read.

Response 3:

We checked and revised them.

Editor

Response 1:

We checked and revised them.

3. Thanks for stating that you'll provide the Tweet IDs upon accept.

Response 2:

We attached the IDs of tweets and retweets as supporting information.

---

## [Decision Letter · Decision Letter 1]

8 Mar 2022

Impact of correcting misinformation on social disruption

PONE-D-21-18422R1

Dear Dr. Iizuka,

We’re pleased to inform you that your manuscript has been judged scientifically suitable for publication and will be formally accepted for publication once it meets all outstanding technical requirements.

Kind regards,

Barbara Guidi

Academic Editor

PLOS ONE

Additional Editor Comments (optional):

Reviewers' comments:

Reviewer's Responses to Questions

**Comments to the Author**

1. If the authors have adequately addressed your comments raised in a previous round of review and you feel that this manuscript is now acceptable for publication, you may indicate that here to bypass the “Comments to the Author” section, enter your conflict of interest statement in the “Confidential to Editor” section, and submit your "Accept" recommendation.

Reviewer #1: All comments have been addressed

Reviewer #3: All comments have been addressed

2. Is the manuscript technically sound, and do the data support the conclusions?

Reviewer #1: Yes

Reviewer #3: Yes

3. Has the statistical analysis been performed appropriately and rigorously? 

Reviewer #1: Yes

Reviewer #3: Yes

4. Have the authors made all data underlying the findings in their manuscript fully available?

Reviewer #1: Yes

Reviewer #3: No

5. Is the manuscript presented in an intelligible fashion and written in standard English?

Reviewer #1: Yes

Reviewer #3: Yes

6. Review Comments to the Author

Reviewer #1: I have gone through the revised version of the manuscript entitled "Impact of correcting misinformation on social disruption". All the comments raised have been attended to by the authors.

Reviewer #3: This paper is well written and well organized. Good work. The results are clearer, but please check the size of the tables because those were very long and maybe creating a problem to keep the paper into proper format.

7. PLOS authors have the option to publish the peer review history of their article (what does this mean?). If published, this will include your full peer review and any attached files.

Reviewer #1: No

Reviewer #3: **Yes: **Krishna Mohan Kudiri

---

## [Editor Report · Acceptance letter]

25 Mar 2022

PONE-D-21-18422R1 

Impact of correcting misinformation on social disruption 

Dear Dr. Iizuka:

I'm pleased to inform you that your manuscript has been deemed suitable for publication in PLOS ONE. Congratulations! Your manuscript is now with our production department. 

Kind regards, 

on behalf of

Dr. Barbara Guidi 

Academic Editor

PLOS ONE